# Thickness Measurement of Endothelium-Descemet Membrane in Descemt Membrane Detachment Patients Using High-Definition Optical Coherence Tomography

**DOI:** 10.3390/jcm11061534

**Published:** 2022-03-11

**Authors:** Wei Wang, Lingjuan Xu, Guanyu Su, Ban Luo, Jing Gao, Yongyao Tan, Hong Zhang, Guigang Li

**Affiliations:** Department of Ophthalmology, Tongji Hospital, Tongji Medical College, Huazhong University of Science and Technology, Wuhan 430030, China; 2011tj0576@hust.edu.cn (W.W.); 2012tj0507@hust.edu.cn (L.X.); d202182014@hust.edu.cn (G.S.); 2009tj0524@hust.edu.cn (B.L.); 2015tj0692@hust.edu.cn (J.G.); m201975980@hust.edu.cn (Y.T.); 1983tj0592@hust.edu.cn (H.Z.)

**Keywords:** cornea, endothelium-Descemet membrane, thickness

## Abstract

Purpose: (1) To measure the corneal endothelium-Descemet membrane (EDM) layer thickness in Descemet membrane detachment (DMD) patients in vivo using high-definition optical coherence tomography (HD-OCT), and to investigate its correlation with age. (2) To explore whether the detachment time will affect the EDM thickness. (3) To explore whether the EDM thickness of cornea with DMD was different from that without DMD. Participants: Patients with DMD were divided into three groups. Group 1 included twenty-three patients whose Descemet membrane (DM) was partial or complete detached from the corneal stroma after various ocular surgeries. Group 2 included eight patients from group 1 who underwent twice HD-OCT examination on different days before the DM reattached to the stroma. Group 3 included nine patients from group 1 who had clear grayscale boundary between the DM and stroma in HD-OCT images after DM reattachment. Methods: All patients underwent HD-OCT and EDM thickness was measured using Image -Pro Plus 6.0. In Group 1, regression analyses were used to evaluate the correlation between EDM thickness and age, and the thickness difference between the ≤50-year-old group and the >50-year-old group was analyzed by independent sample *t*-test. In Group 2, paired samples *t*-test was used to check whether detachment time would affect EDM thickness. In Group 3, paired samples *t*-test was used to check whether the EDM thickness of cornea with DMD was different from that without DMD. *p* < 0.05 was considered significant. Results: In Group 1, the EDM thickness measured on the first post-operative day was 27.8 ± 3.6 μm, and a positive correlation was found between EDM thickness and age (r = 0.619, *p* < 0.05). The EDM thickness of ≤50-year-old group and >50-year-old group were 23.9 ± 3.2 and 29.2 ± 2.6 μm, and there was a significant difference between the two groups (*p* = 0.001). In Group 2, the first measurement of EDM thickness was 27.5 ± 4.0 μm, the second measurement was 27.6 ± 4.2 μm, the interval between the two measurements was 2.1 ± 1.6 days, and there was no significant difference between the two measurements (*p* = 0.328). In Group 3, the EDM thickness with DM detachment was 28.3 ± 3.5 μm, with DM reattachment was 23.4 ± 2.4 μm, there was a significant difference between the two measurements (*p* = 0.002). Conclusions: The EDM thickness in the state of DMD is thicker than its actual thickness in normal cornea, and EDM thickness of the >50-year-old group is much thicker than that of the ≤50-year-old group.

## 1. Introduction

Descemet membrane endothelial keratoplasty (DMEK) is one of the main methods of endothelial transplantation, compared with Descemet stripping endothelial keratoplasty (DSEK), it provides faster visual recovery, minimal refractive changes and a lower rejection rate [1,2,3,4]. DSEK implants healthy donor endothelium, Descemet membrane, and posterior stroma, while DMEK eliminates the donor stromal layer. The graft of DSEK is thicker than DMEK; therefore, it is less difficult to operate, but it will lead to postoperative hyperopic shift [5]. DMEK is more surgically challenging, especially the difficulty of donor insertion and attachment, the key reason is the endothelium-Descemet membrane (EDM) graft is too thin. Price et al. [6] have described that the EDM grafts which detached from the corneal stroma will naturally fold in the “roll shape”, while older donor grafts are easier to unfold in the recipient’s anterior chamber. Therefore, doctors tend to choose elderly corneal donors (>50 years old) to make EDM grafts in DMEK [4,6,7]. Some surgeons even prefer corneas from donors older than 55 years [8,9]. Some studies have found that the higher donor age seems to be associated with lower chances of graft detachment after DMEK [10].

Why are the EDM grafts from elderly donors easier to unfold than young donors? We infer that the EDM thickness may be the key to the above problem. However, to our knowledge, there are few relevant clinical studies supporting this hypothesis. This study will demonstrate the above hypothesis by measuring the EDM thickness in vivo.

Although some research institutions use ultra-high-resolution optical coherence tomography (UHR-OCT) with 0.95 μm axial resolution to image the Descemet membrane (DM) and measure its thickness in vivo [11], the commercially available time-domain or spectral-domain AS-OCT instruments have an axial resolution of only 11–18 μm which is insufficient to distinguish the boundary between the corneal stroma and EDM in normal cornea. The EDM grafts in DMEK are actually in the Descemet membrane detachment (DMD) state, which boundary between the corneal stroma and EDM are clear in AS-OCT images. Therefore, we can select DMD patients with various surgery causes, measure their EDM thickness in vivo and analyze the correlation with age. At the same time, in order to improve the measurement accuracy, this study used high-definition optical coherence tomography (HD-OCT) with 5 μm axial resolution to depict the boundary of EDM in vivo and measured its thickness. HD-OCT is an accurate and objective method for corneal measurement. Wertheimer et al. [12] use HD-OCT to measure the corneal optical density as an objective value for corneal changes in Fuchs endothelial corneal dystrophy. Keidel et al. [13] use HD-OCT to assess the corneal cystine crystal deposition and found it’s an objective quantification. Syed et al. [14] use HD-OCT to measure the EDM thickness of eye bank donor corneas and found the mean thickness was 17 ± 4 μm.

## 2. Subjects

This study has been approved by the ethics committee of Tongji Hospital of Tongji Medical College of Huazhong University of Science and Technology, and written consent was collected from each patient, and was carried out in compliance with the tenets of the Declaration of Helsinki.

The selected subjects are from the inpatients who were treated in the corneal disease group of Tongji Hospital of Tongji Medical College of Huazhong University of Science and Technology from January 2016 to December 2018. Group 1 included 23 DMD patients whose EDM were partial or complete detached from the corneal stroma after various surgery, including 3 patients after phacoemulsification (phaco), 2 patients after penetrating keratoplasty (PKP), 2 patients after descemetic deep anterior lamellar keratoplasty (dDALK, with Descemet membrane-endothelium was exposed), 12 patients after predescemetic deep anterior lamellar keratoplasty (pdDALK, with minimal residual stroma was left behind), and 4 patients after DMEK. The patients age ranged from 16 to 75 years (for PKP and DMEK cases, donor age was used), mean age 55.2 ± 14.0 years. All patients were imaged by HD-OCT examinations (Carl Zeiss HD-OCT 5000, Zeiss, Germany) on the first post-operative day. Group 2 included 8 patients from Group 1 who underwent a second HD-OCT examination before the DM reattached to the stroma, and the interval between the two HD-OCT examinations was 1–5 days (mean 2.1 ± 1.6 days). All 23 patients had DM reattachment spontaneously or by air injection to the anterior chamber, except 1 DMEK patient and 1 phaco patient lost follow-up, the remaining 21 patients took more than 3 months follow-up after DM reattachment and received HD-OCT examinations at the third month. Group 3 included 9 patients from Group 1 who had a clear grayscale boundary between the corneal stroma and DM in HD-OCT images after DM reattachment (Figure 1 and Figure 2).

## 3. Image Acquisition and Processing

Through the manual measurement mode of Carl Zeiss HD-OCT 5000, the scale of 30 μm was marked on the original OCT image as a reference for later measurement. The image was saved in JPG format and then imported into Image—Pro Plus 6.0 (MediaCybernetics, Silver Spring, MD, USA). The measurement mode was selected, and the scale was defined. The scale in the software was overlapped with the scale in the original image to determine the unit. The measurement was performed by two independent technicians. The selection criteria of measurement points were as follows: 1. Select 4 measuring points in the area where the EDM was separated from the stroma and try to distribute them evenly; 2. The boundary of the EDM of the measuring points should be clear and the front and rear boundaries should be parallel as far as possible, and enlarge the image to determine the boundary of EDM artificially, then the software automatically calculate the length. The data were recorded and the average value was calculated. Because the literature found that “the thickness of the DM and endothelium layers was consistent over the entire region of the cornea, there is no significant difference between the central and peripheral areas of the cornea” [11], in this study we only selected the measurement points in the area of DMD. Although the area and range of the DMD were different in different patients, we believe that this would not affect the results. The analysis was performed in a masked fashion, and all of the samples were analyzed by one person to minimize bias.

## 4. Statistical Analysis

For statistical evaluation of the data, SPSS 19.0 (SPSS Inc., Chicago, IL, USA) was used. In Group 1, a linear regression was used to analyze the relationship between EDM thickness and age, and an independent sample *t*-test was used to check the differences of the EDM thickness between the ≤50-year-old group and >50-year-old group. In Group 2, a paired samples *t*-test was used to check whether the detachment time will affect the EDM thickness. In Group 3, a paired samples *t*-test was used to check whether the thickness of EDM in the state of DMD is different from that of the normal cornea. *p* < 0.05 was considered significant. Values are presented as means ± standard deviation.

In Group 1, regression analyses were used to evaluate the correlation between EDM thickness and age, and the thickness difference between the ≤50-year-old group and the >50-year-old group was analyzed by independent sample *t*-test. In Group 2, a paired samples *t*-test was used to check whether detachment time would affect EDM thickness. In Group 3, a paired samples *t*-test was used to check whether the EDM thickness of the cornea with DMD was different from that without DMD. *p* < 0.05 was considered significant.

## 5. Results

For all 23 subjects in Group 1, the thickness of the EDM measured on the first post-operative day was 27.8 ± 3.6 μm (range, 19.4–34.8, Figure 3), and a positive correlation was found between EDM thickness and age (r = 0.619, *p* < 0.05, Figure 4). The EDM thickness of the ≤50-year-old group was 23.9 ± 3.2 μm (range, 19.4–27.4), for the >50-year-old group was 29.2 ± 2.6 μm (range, 24.2–34.8), and there was a significant difference between the two groups (*p* = 0.001, Table 1).

Data are expressed as mean ± standard deviation, dDALK descemetic deep anterior lamellar keratoplasty, pdDALK predescemetic deep anterior lamellar keratoplasty, PKP penetrating keratoplasty, DMEK descemet membrane endothelial keratoplasty, phaco phacoemulsification

In Group 2, the 8 subjects who had two HD-OCT examinations before DM reattachment (the interval between the two examinations was 1–5 days, with an average of 2.1 ± 1.6 days, Table 2), the age range (for PKP and DMEK cases, the donor age was used) was from 16 to 73 year (average 51.4 ± 20.2 year). The first measurement of EDM thickness was 27.5 ± 4.0 μm (range, 19.4–30.5), the second measurement was 27.6 ± 4.2 μm (range, 19.4–30.9), and there was no significant difference between the two measurements (*t* = −1.051, *p* = 0.328).

In Group 3, the 9 subjects whose boundary between the corneal stroma and DM were clear in HD-OCT images after DM reattachment (Table 3), the age range (for PKP and DMEK cases, donor age was used) was from 16 to 75 year (average 58.9 ± 17.6 year). The EDM thickness with DM detachment was 28.3 ± 3.5 μm (range, 24.4–34.8), the EDM thickness with DM reattachment was 23.4 ± 2.4 μm (range, 18.3–30.3), and there was a significant difference between the two measurements (*t* = 4.407, *p* = 0.002).

## 6. Discussion

How to reduce the difficulty of DMEK has always been the research focus of corneal transplantation experts [15,16]. In particular, the depth of anterior chamber of Chinese people is shallow [17]. How to reduce the operation steps in the limited anterior chamber space and how to unfold the EDM graft as soon as possible are the direction that DMEK experts are trying to explore [18]. In clinical practice, we found that EDM grafts from elderly donors are easy to separate from the donor corneal stroma and easier to unfold in the recipient’s anterior chamber. Therefore, we tend to choose the elderly donor over 50 years old to make the graft. There are also some literatures that put forward the same clinical experience [6,7,8,9]. In this study, we hope to explain this clinical experience. Therefore, we used HD-OCT to measure the thickness of EDM in DMD in vivo, and analyzed its correlation with age. Our data showed a strong positive correlation between age and EDM thickness in DMD state, indicating that the EDM thickness increases with age. In this study, we found that the EDM thickness of the ≤50-year-old group was 23.9 ± 3.2 μm (range, 19.4–27.4), for the >50-year-old group was 29.2 ± 2.6 μm (range, 24.2–34.8), and there was a significant difference between the two groups. This result also supports our previous hypothesis that the thickness maybe the reason why grafts of the elderly EDM donors are easier to unfold in the anterior chamber of the recipient in DMEK. For DMEK, older donors’ grafts are relatively thicker, which are easier to unfold. So less and simpler surgical procedures are the greatest protection for endothelial cells. Therefore, for DMEK beginners, based on the consideration of the thickness of EDM grafts, selecting elderly donors (>50 years old) to make grafts will help to reduce the difficulty and improve the success rate of surgery.

In the early studies, some researchers used transmission electron microscope to measure the thickness of DM in vitro. Murphy et al. [19] measured the mean DM thickness was 10.07 ± 0.99 μm in the ages ranging from 32 to 44 years (*n* = 6) and 11.61 ± 2.32 μm in the ages ranging from 55 to 68 years (*n* = 10), Johnson et al. [20] measured the mean DM thickness was 10.88 ± 2.49 μm in the ages ranging from 52 to 68 years (*n* = 6). With the progress of OCT technology, some studies have begun to measure DM thickness in vivo. Shousha et al. [21] use UHR-OCT with 3 μm axial resolution to measure the mean central thicknesses of DM in normal young people (20 eyes of 13 young subjects, age range from 19 to 44 years) was 10 ± 3 μm, in normal elderly people (20 eyes of 15 elderly patients, age range from 60 to 86 years) was 16 ± 2 μm, and found there was a significant correlation between age and DM thickness. Bizheva et al. [11] use UHR-OCT with 0.95 μm axial resolution to measure the thickness of DM and endothelium layer of normal cornea in vivo (20 subjects with ages varying from 20 to 60 years), found the mean thickness of DM was 10.4 ± 2.9 μm, endothelium layer was 4.8 ± 0.4 μm. In our study, the mean EDM thickness in DMD state was 27.8 ± 3.6 μm (range, 19.4–34.8). Compared with the DM thickness reported in previous studies, even considering the thickness of corneal endothelium, the EDM thickness in DMD state is significantly thicker than that in normal cornea. In order to further verify the above point of view, in Group 3 of this study, we found the EDM thickness of DM detachment was 28.3 ± 3.5 μm, the EDM thickness of DM reattachment was 23.4 ± 2.4 μm, and there was a significant difference between the two measurements (*p* = 0.002). Therefore, the question arises: since the thickness of EDM will increase after detachment from corneal stroma, will the detachment time affect its thickness? In Group 2 of this study, we collected 8 subjects who underwent two HD-OCT examinations before DM reattachment (the interval between the two examinations was 1–5 days), the first measurement of EDM thickness was 27.5 ± 4.0 μm, the second measurement was 27.6 ± 4.2 μm, and there was no significant difference between the two measurements (*p* = 0.328). In conclusion, although the EDM thickness may increase in DMD state, the degree of thickening is limited; within the measurement accuracy range of this study, the thickness of EDM did not change significantly within 1–6 days after EDM detachment.

The view of the pre-Descemet layer (PDL, also called Dua’s layer) is increasingly accepted [22]. Dua et al. [23] had classified three types of DMD in OCT: The type 1 (the PDL and DM were detached together) is seen as a characteristic straight line, like the chord of a circle. The type 2 (only DM was detached) is undulating and is made of 2 parallel hyperreflective lines separated by a narrow dark space. The type 3 (the PDL and DM were detached but also separated from each other) has 2 components, the anterior taut hyperreflective line like a chord of a circle representing the PDL separated from the posterior stroma, and the posterior straight or undulating double contour line representing the DM, with the latter also separated from the former. We reviewed the OCT images of all our subjects and found all the patients’ DMD were type 2. We speculate that the PDL detachment was rarely observed may be related to race and surgery, or it may be related to the small number of cases we observed.

To our knowledge, this study is the first observation that the thickness of EDM will increase to a limited extent when it is detached from the corneal stroma. In addition to the change of EDM thickness caused by postoperative inflammation of corneal endothelial cells, is there the possibility of edema in DM layer? We can see the clinical signs of “DM folds” in some corneal disease, but few studies use the description of “DM edema” in the literature. Yu et al. [24] believed the DM folds after cataract surgery are caused by the acute edema of the corneal stroma which could push DM posteriorly and forming a wavy DM layer. The cause of corneal stroma edema and thickening is that the hydration of the proteoglycans such as chondroitin sulphate between the corneal collagen fibers which causes the interfibrillar spacing increase [25]. DM is composed of collagen components, such as VI and VIII, and noncollagenous components, including laminin, fibronectin, heparin sulphate, dermatan sulphate, and *p* component [26,27,28]. We speculate that the DM may be similar to the corneal stroma, the DM thickens through water absorption and expansion of polysaccharide between collagen fibers. However, due to the dense arrangement of collagen fibers in DM, the degree of this thickening is limited. This can explain that the thickness of EDM does not change significantly within 1–6 days after EDM detachment. In view of the fact that EDM is a very thin layer, it may thicken gradually as the time of EDM detaching from the corneal stroma increases, but within the measurement accuracy of this study, we cannot observe the change of EDM thickness in a short time. This requires the use of HD-OCT with higher resolution in follow-up studies, while increasing the sample size to further improve the observation.

The main limitation of this study is that we used DMD patients to measure the thickness of EDM, the samples of three groups are small, because there are not many patients with DMD in clinical, especially young samples. When we analyzed our OCT images, we had found some artifacts such as Figure 2B. We analyzed the reason and thought it was caused by the coincidence of the reflective point of the light source in the cornea and the center point of the lens at a special angle. This artifact is not caused by the patient’s eye movement; therefore, it will not affect the EDM measurement.

## 7. Conclusions

This study analyzed from an anatomical perspective that DM changes with age may be the reason why grafts of the elderly EDM donors are easier to unfold in the recipient’s anterior chamber, and explained why doctors recommend selecting older donors (>50 years old) to make EDM grafts in DMEK. Additionally, the thickness of EDM may increase due to edema and other factors after it is detached from the corneal stroma. However, the extent of this thickening is limited, within the measurement accuracy of this study, the thickness of EDM did not change significantly within 1–6 days after EDM detachment.

## Figures and Tables

**Figure 1 jcm-11-01534-f001:**
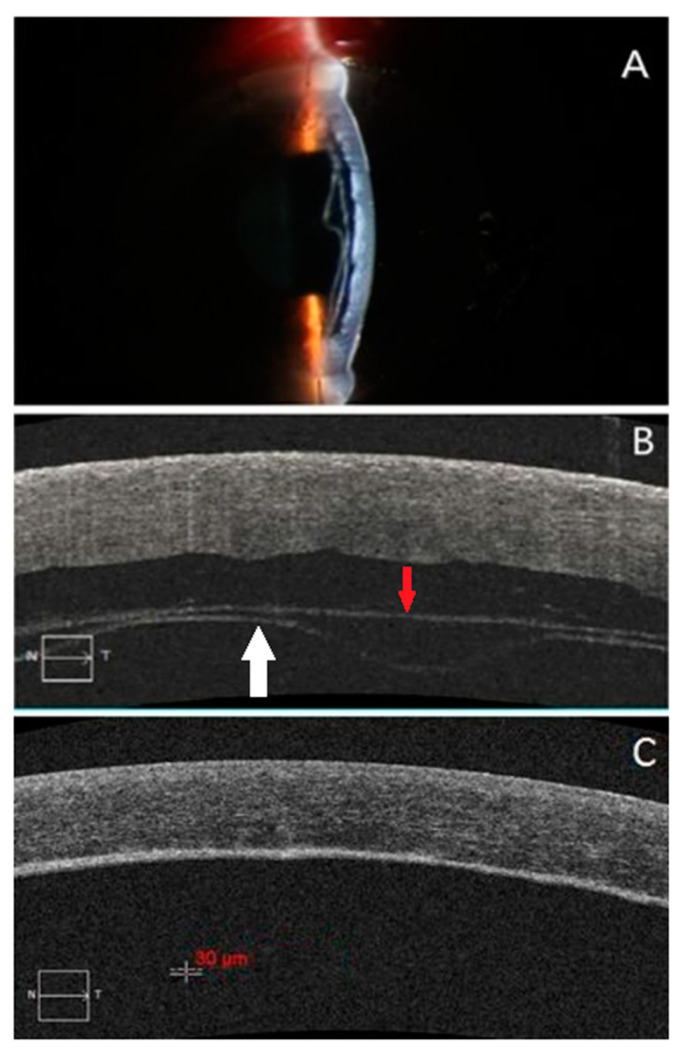
The boundary between the corneal stroma and EDM in HD-OCT images are unclear after DM reattachment in pdDALK patients. (**A**): One case of a 59-year-old male with fungal keratitis and underwent pdDALK in his left eye, triple chamber was observed on the first post-operative day. (**B**): The HD-OCT image shows the graft separated from the residual stromal bed, and the EDM layer (white arrow) was detached from the stroma (red arrow). The patient underwent the interface drainage along with descemetopexy. (**C**): 3 months later the HD-OCT image shows that the grayscale boundary between the graft and the residual stromal bed was clear, but there was no obvious boundary between the residual stroma and EDM layer which is difficult to measure the EDM thickness.

**Figure 2 jcm-11-01534-f002:**
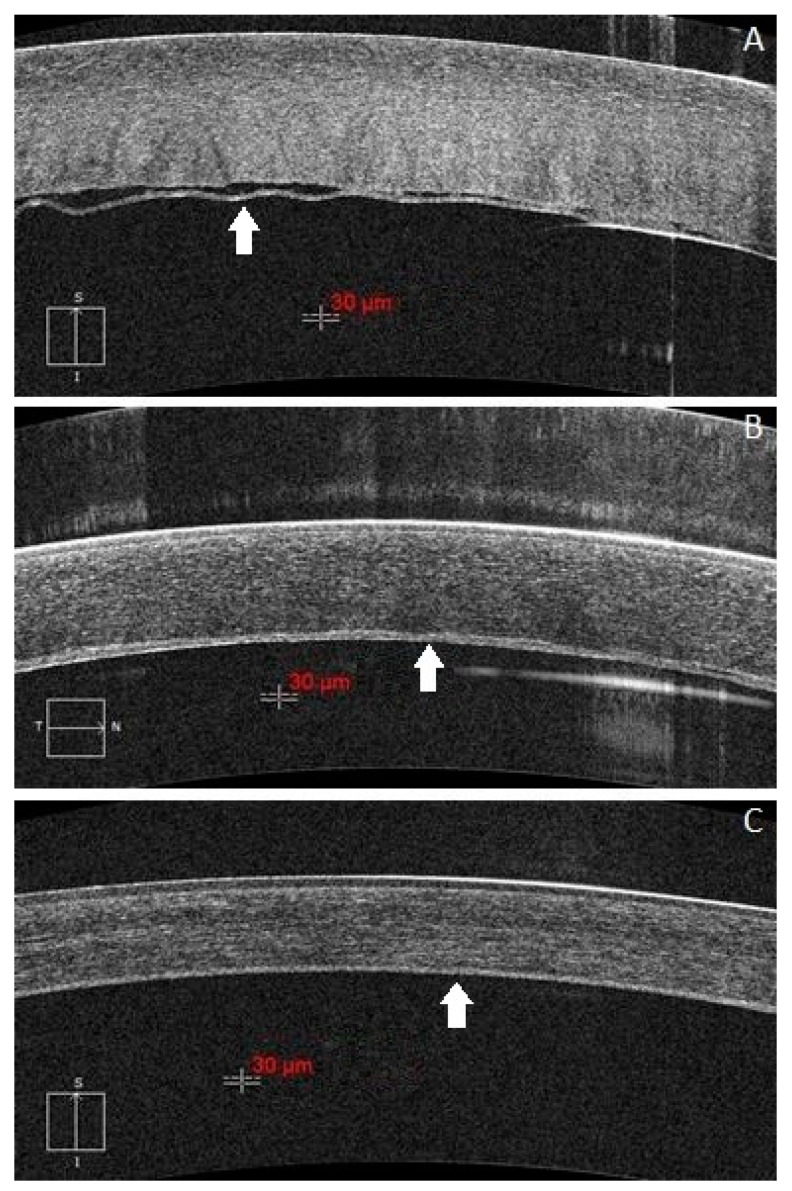
The boundary between the corneal stroma and EDM in HD-OCT images are clear after DM reattachment in dDALK patients. (**A**): One case of a 16-year-old male with keratoconus and underwent dDALK in his right eye, the EDM (white arrow) was partial separated from the graft on the first post-operative day. (**B**): The DM reattached spontaneously, but the thickness of EDM was uneven due to inflammation at 1 month after DM reattachment. (**C**): The inflammation has subsided and the thickness of EDM was stable and clear for measurement at 3 months after DM reattachment.

**Figure 3 jcm-11-01534-f003:**
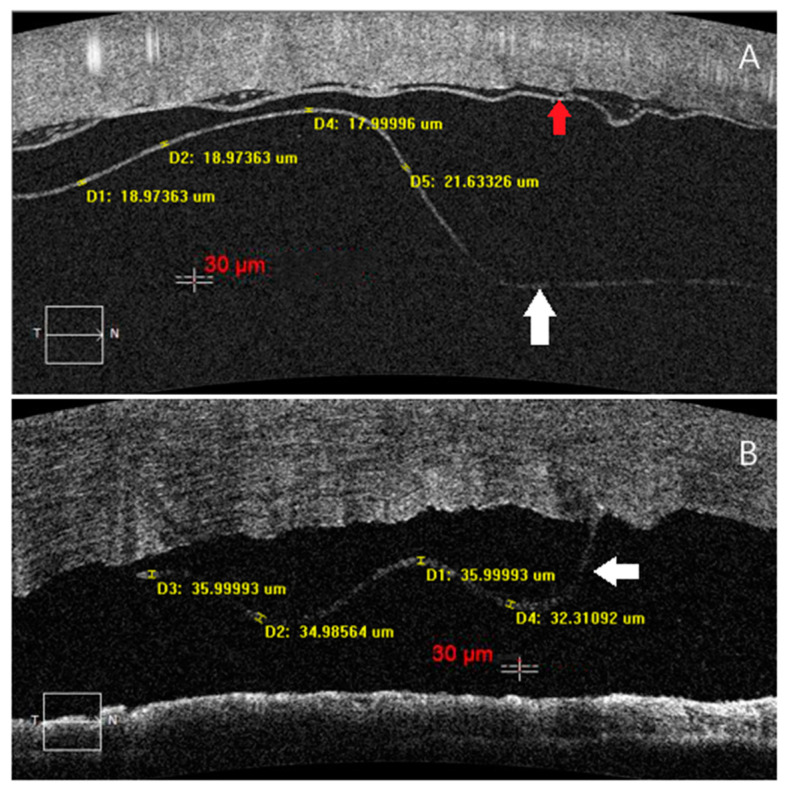
(**A**): One case of a 25-year-old male with cornea leukoma in his right eye, who underwent pdDALK with little residual stroma was left behind, the EDM layer (white arrow) was observed separated from the residual stroma bed (red arrow) by HD-OCT on the first post-operative day, and it’s the thinnest EDM in all subjects. (**B**): One case of a 53-year-old male with bacterial keratitis in his right eye, who underwent PKP (the donor age was 75 years), the EDM layer was observed partial separated from stroma by HD-OCT on the first post-operative day, and it was the thickest EDM in all of the subjects.

**Figure 4 jcm-11-01534-f004:**
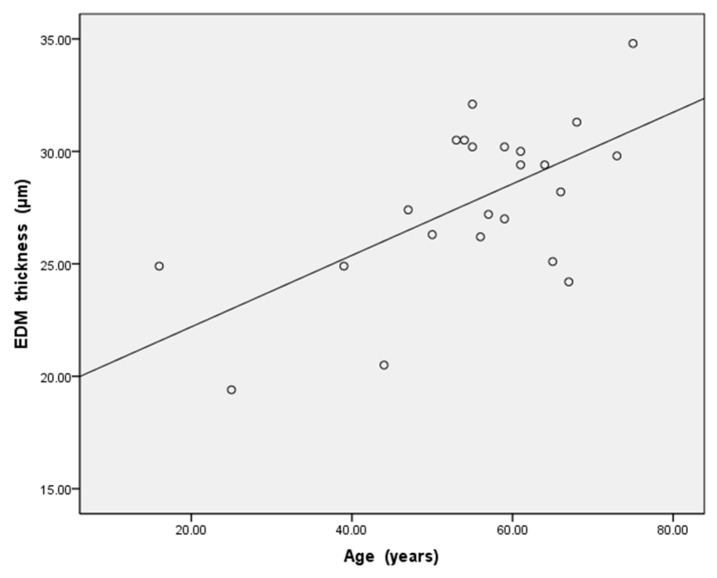
In all 23 subjects of Group1, the thickness of the EDM tended to become thicker with age. There was a highly significant positive correlation (r = 0.619, *p* < 0.05) between the EDM thickness and age.

**Table 1 jcm-11-01534-t001:** Demographic data in the ≤50-year-old and >50-year-old groups.

		≤50-Year-Old *n* = 6	>50-Year-Old *n* = 17	*p* Value
Age (years)		36.8 ± 13.5	61.6 ± 6.6	0.015
the EDM thickness on the first post-operative day (μm)		23.9 ± 3.2	29.2 ± 2.6	0.001
the DMD cause	dDALK	1	1	
	pdDALK	5	7	
	PKP	0	2	
	DMEK	0	4	
	phaco	0	3	

**Table 2 jcm-11-01534-t002:** Data of 8 subjects who had two HD-OCT examinations before DM reattachment.

No.	Age (year)	The EDM Thickness on the First Examination (μm)	The EDM Thickness on the Second Examination (μm)	The Interval between Two Examinations (day)	The DMD Cause
1	16	24.9	25.1	1	dDALK
2	25	19.4	19.4	4	pdDALK
7	53 !	30.5	30.9	5	PKP
8	54	30.5	30.7	1	pdDALK
15	61	30.0	30.3	1	dDALK
17	64 !	29.4	29.5	1	DMEK
18	65 !	25.1	24.4	3	DMEK
22	73	29.8	30.5	1	phcao

! For PKP and DMEK cases, the donor age was used.

**Table 3 jcm-11-01534-t003:** Data of 9 subjects whose boundary between the corneal stroma and DM were clear in HD-OCT images after DM reattachment.

No.	Age (year)	The EDM Thickness with DM Detachment (μm)	The EDM Thickness with DM Reattachment (μm)	The DMD Cause
1	16	24.9	24.2	dDALK
7	53 !	30.5	27	PKP
11	56 !	26.2	24.2	DMEK
15	61	30.0	24.4	dDALK
17	64 !	29.4	24.2	DMEK
18	65 !	25.1	18.3	DMEK
20	67	24.2	21.4	phaco
22	73	29.8	24.5	phaco
23	75 !	34.8	22.7	PKP

! For PKP and DMEK cases, the donor age was used.

## Data Availability

All data generated or used during the study appear in the submitted article.

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
