# Peer review of "Thickness Measurement of Endothelium-Descemet Membrane in Descemt Membrane Detachment Patients Using High-Definition Optical Coherence Tomography"

_jcm, 2022, doi:10.3390/jcm11061534_

Round 1

Reviewer 1 Report

Review of JCM 1586595
Title: Use of high-density optical coherence tomography to measure the endothelium-Descemet membrane thickness in Descemet membrane detachment

Authors: W. Wang et al.

Authors have reported a clinical anterior segment OCT (ASOCT) study to investigate thickness of endothelium-Descemet membrane (EDM or called as DM) below stromal layer of cornea, started from one curiosity that why EMDs from older donors are more unfolding compared to those from young donors for endothelial transplantation. Authors’ inference was because the EMDs from the older people are the thicker. To measure the thickness of EDM, authors’ have selected and imaged EDMs detached from the patients following the different types of transplantation surgeries using a commercial ASOCT. The results have confirmed Authors’ hypothesis that the EDM is gradually thicker for age.

As authors’ mentioned, measuring EDM’s thickness using OCT itself is not new, but it seems to be interesting to reveal correlation between the EDM thickness and age using moderate resolution OCT system in clinic, hinting that the graft of elderly EDM donors is less rollable and then preferred to be used for cornea surgery. Manuscript is quite well written in English, and describes the procedure and methods for study in detail. The experimental results including OCT images are clear and evident. Tables are clear to see as well. Therefore, manuscript could be accepted given few comments would be addressed.

  1. Please write the number of IRB protocol approved for this study.
  2. The thickness measurements were conducted using image software, which is subjective for setting points in the area. How about using line profiles across the EDM? Authors would pick up the two peaks from the two boundaries and then calculate the pixels between them, that can be converted to the physical length.
  3. According to a recent ASOCT paper, HS Dua et al, “Descemet membrane detachment: a novel concept in diagnosis and classification,” American Journal of Ophthalmology 218, 84-98 (2020), they have revealed the existence of pre-Descemet layer (PDL) dubbed as Dua-Fine layer, another dimension to DM. More interestingly, they have classified types of Descemet membrane detachment: type 1, where the PDL and DM were detached together; type 2, where only the DM was detached; and mixed (type 3), where the PDL and DM were detached but also separated from each other. Considering their classification, in your work, Fig 3A would be involved into type 3 (PDL and DM separated each other), however, Fig. 3B shows type 2 case (PDL and DM attached each other). Please see that Fig 3A and Fig 3B in authors’ result are quite similar to Fig. 2(bottom) and Fig. 2 (middle) respectively in the Dua’s paper. In that way, I am concerned that you should have considered both PDL and DM for EDM thickness measurements, especially for Fig. 3A case where you have seemed to measure the thickness of only DM separated from PDL.
  4. What is meaning of 'high-density' of high density OCT? Is it different from 'high-resolution'?

Author Response

Response to Reviewer 1 Comments

Authors have reported a clinical anterior segment OCT (ASOCT) study to investigate thickness of endothelium-Descemet membrane (EDM or called as DM) below stromal layer of cornea, started from one curiosity that why EMDs from older donors are more unfolding compared to those from young donors for endothelial transplantation. Authors’ inference was because the EMDs from the older people are the thicker. To measure the thickness of EDM, authors’ have selected and imaged EDMs detached from the patients following the different types of transplantation surgeries using a commercial ASOCT. The results have confirmed Authors’ hypothesis that the EDM is gradually thicker for age.

As authors’ mentioned, measuring EDM’s thickness using OCT itself is not new, but it seems to be interesting to reveal correlation between the EDM thickness and age using moderate resolution OCT system in clinic, hinting that the graft of elderly EDM donors is less rollable and then preferred to be used for cornea surgery. Manuscript is quite well written in English, and describes the procedure and methods for study in detail. The experimental results including OCT images are clear and evident. Tables are clear to see as well. Therefore, manuscript could be accepted given few comments would be addressed.

Point 1: Please write the number of IRB protocol approved for this study.

Response 1: Thank you for your suggestion. This study is a retrospective study, not a clinical trial. Because it involves the examination results of patients with keratoplasty, the research protocol has been approved by the ethics committee and written consent was collect from each patient, but there is no IRB protocol.

Point 2: The thickness measurements were conducted using image software, which is subjective for setting points in the area. How about using line profiles across the EDM? Authors would pick up the two peaks from the two boundaries and then calculate the pixels between them, that can be converted to the physical length.

Response 2: Response 2: Thank you for your suggestion. Using line profiles is a good method to measure EDM thickness more objectively. We reviewed the OCT images of all our patients and found that some patients' EDM did not have continuous and clearly anterior and posterior boundary due to postoperative corneal edema, as shown in the figure. Therefore, using line profiles may increase measurement difficulty. In follow-up studies, as higher-resolution OCT becomes available clinically, we will use the line profiles method to make the measurement more precise and objective.(Please see the supplymentary figure in the word document that i submited)

Point 3: According to a recent ASOCT paper, HS Dua et al, “Descemet membrane detachment: a novel concept in diagnosis and classification,” American Journal of Ophthalmology 218, 84-98 (2020), they have revealed the existence of pre-Descemet layer (PDL) dubbed as Dua-Fine layer, another dimension to DM. More interestingly, they have classified types of Descemet membrane detachment: type 1, where the PDL and DM were detached together; type 2, where only the DM was detached; and mixed (type 3), where the PDL and DM were detached but also separated from each other. Considering their classification, in your work, Fig 3A would be involved into type 3 (PDL and DM separated each other), however, Fig. 3B shows type 2 case (PDL and DM attached each other). Please see that Fig 3A and Fig 3B in authors’ result are quite similar to Fig. 2(bottom) and Fig. 2 (middle) respectively in the Dua’s paper. In that way, I am concerned that you should have considered both PDL and DM for EDM thickness measurements, especially for Fig. 3A case where you have seemed to measure the thickness of only DM separated from PDL.

Response 3: Response 3: Thank you very much for providing Dua’s paper to help us have a deeper understanding of DMD, and we think the classification of DMD will have a very positive impact on helping cornea surgeons re-understand and diagnose DMD. In Dua’s paper, AS-OCT of DMD showed 3 patterns: The type 1 DMD (the PDL and DM were detached together) is seen as a characteristic straight line, like the chord of a circle. The type 2 DMD (only the DM was detached) is undulating and is made of 2 parallel hyperreflective lines separated by a narrow dark space. The type 3 DMD (the PDL and DM were detached but also separated from each other) has 2 components, the anterior taut hyperreflective line like a chord of a circle representing the PDL separated from the posterior stroma, and the posterior straight or undulating double contour line representing the DM, with the latter also separated from the former. In our work, we think Fig 3A is the type 2 DMD, not the type 3, because the middle layer is an undulating line, not a characteristic straight line. We reviewed the case of Fig 3A and the operation video, and we confirmed that the operation performed by the surgeon was predescemetic deep anterior lamellar keratoplasty, with some residual stroma was left. In Fig 3A, the middle layer is thicker than the bottom layer, which also reflects that the middle layer contains residual stroma and PDL. However, we will add the discussion on DMD classification. We reviewed OCT images of all our patients and found they were all type 2, so we only calculated DM and endothelial thickness when measuring EDM. Thank you again for your suggestion to consider PDL when making measurements, we will pay more attention to this in follow-up studies.

Point 4: What is meaning of 'high-density' of high density OCT? Is it different from 'high-resolution'?

Response 4: : Thank you for pointing out our mistakes. Yes, it should be 'high-resolution’, we have corrected it in the manuscript.

Reviewer 2 Report

Overall, the study seems to be interesting and the application of using OCT for Endothelium–Descemet Membrane thickness measurement to correlate and assess  EDM degradation in accordance to age is of general interest to readers in the same field of research. However, the manuscript should be improved as per the suggestions to consider for publications. The result analysis is extensive, this will be useful for understanding the main aspect of the study.

Comments:

  1. The Introduction and background is too less and must be improved. Multiple research groups have used other functional OCT imaging methods to analyze internal layers of the retina and cornea in the eye. These should be included with relevant citations.
  2. Fig 1-3, individual layers should be labeled, the scale bar is too small.
  3. 2 (B), the presence of image ghosting by reflection should be explained and its potential cause of occurrence should be discussed.
  4. The authors should consider including a Volume and en face visualization of the Descemet membrane. This can be achieved by carefully cropping out the rest of the layers. Providing this will help readers to fully visualize the difference between control and treated subjects.

Author Response

Response to Reviewer 2 Comments

Overall, the study seems to be interesting and the application of using OCT for Endothelium–Descemet Membrane thickness measurement to correlate and assess  EDM degradation in accordance to age is of general interest to readers in the same field of research. However, the manuscript should be improved as per the suggestions to consider for publications. The result analysis is extensive, this will be useful for understanding the main aspect of the study.

Point 1: The Introduction and background is too less and must be improved. Multiple research groups have used other functional OCT imaging methods to analyze internal layers of the retina and cornea in the eye. These should be included with relevant citations.

Response 1: Thank you for your suggestion. We have added relevant content and citations in the introduction and background.

Point 2: Fig 1-3, individual layers should be labeled, the scale bar is too small.

Response 2: Thank you for your suggestion. We have added labels to each layer in the Fig 1-3. Because the thickness of most EDM is between 20-30 μm, we chose a 30 μm scale in the figure for more intuitive reference.

Point 3: 2 (B), the presence of image ghosting by reflection should be explained and its potential cause of occurrence should be discussed.

Response 3: Thank you for your suggestion. In this study, we used Carl Zeiss HD-OCT 5000, when the machine scans, a beam of light will shine on the cornea, and a reflective point can be seen in the observation screen. When moving the lens to a certain angle, the center point of the lens just coincides with the reflective point on the same axis, which will produce this highly reflective light signal. The better the coincidence, the brighter the reflection, as shown in the figure. This phenomenon is very similar like the light shining on the mirror. If a human eye just looks at the light reflected by the mirror, it will feel very dazzling, but if it looks at the mirror outside the reflection path, the light will not be so dazzling. This artifact is not caused by the patient's eye movement, so it will not affect the EDM measurement. We will avoid situations where this artifact appears in the OCT image acquisition in the follow-up studies. We added an explanation for this artifact in the discussion.(please see the figure in the word document that i submitted)

Point 4: The authors should consider including a Volume and en face visualization of the Descemet membrane. This can be achieved by carefully cropping out the rest of the layers. Providing this will help readers to fully visualize the difference between control and treated subjects.

Response 4: Thank you for your suggestion. This is a very good suggestion for measuring the volume and visualization of DM and we will consider use this method in the subsequent study. However, in this study, it is convenient to observe the extent and degree of detachment by showing the relationship between EDM and corneal stroma. In addition, 12 of our subjects were pdDALK, preserving corneal stroma can better show the relationship between EDM and residual corneal bed.

Reviewer 3 Report

Well-prepared manuscript. Please, review the entire manuscript and clearly check English grammar and language. Descriptive statistical analysis is good. Personally, I think the samples of three groups are too small to state conclusions.

However, I think "elderly EDM donors are more easily unfold in the anterior chamber of the recipient" is an acceptable observation very useful for cornea surgeons, even though the most of them have already known this practical aspect from their routinely surgery.

Author Response

Response to Reviewer 3 Comments

Well-prepared manuscript. Please, review the entire manuscript and clearly check English grammar and language. Descriptive statistical analysis is good. Personally, I think the samples of three groups are too small to state conclusions.

However, I think "elderly EDM donors are more easily unfold in the anterior chamber of the recipient" is an acceptable observation very useful for cornea surgeons, even though the most of them have already known this practical aspect from their routinely surgery.

Response : Thank you for your suggestion. We have carefully examined the manuscript and changed some grammar and language errors. The samples of three groups are small, because there are not many patients with DMD in clinical, especially young samples. We have added relevant explanations in the discussion part, and we will collect more cases in future clinical work, and hope to illustrate the relationship between age and EDM thickness through more sample sizes in the follow-up study.

Round 2

Reviewer 1 Report

I think that authors constructively made reply to reviewer's comments, and their responses to the comments and concerns seem to be properly reflected in the revised manuscript. 

If possible, please consider to re-change the manuscript title to "Thickness measurement of Endothelium-Descemet Membrane in Descemt membrane detachment patients using high-definition Optical Coherence Tomography".

Author Response

Response to Reviewer 2 Comments

I think that authors constructively made reply to reviewer's comments, and their responses to the comments and concerns seem to be properly reflected in the revised manuscript. 

If possible, please consider to re-change the manuscript title to "Thickness measurement of Endothelium-Descemet Membrane in Descemt membrane detachment patients using high-definition Optical Coherence Tomography".

Response:Thank you for your appreciation of this article, and I totally agree with you, the title “Thickness measurement of Endothelium-Descemet Membrane in Descemt membrane detachment patients using high-definition Optical Coherence Tomography” is better. And I have already changed the title in the manuscript
